# Heterogeneity of proteome dynamics between connective tissue phases of adult tendon

Howard Choi[1†], Deborah Simpson[2†], Ding Wang[1], Mark Prescott[2], Andrew A Pitsillides[3], Jayesh Dudhia[4], Peter D Clegg[5], Peipei Ping[1], Chavaunne T Thorpe[3*]

[1]Department of Physiology and Medicine, David Geffen School of Medicine, UCLA, Los Angeles, United States; [2]Centre for Proteome Research, Biosciences Building, Institute of Integrative Biology, University of Liverpool, Liverpool, United Kingdom; [3]Department of Comparative Biomedical Sciences, Royal Veterinary College, London, United Kingdom; [4]Department of Clinical Sciences and Services, Royal Veterinary College, Hatfield, United Kingdom; [5]Department of Musculoskeletal Biology, Institute of Ageing and Chronic Disease, University of Liverpool, Liverpool, United Kingdom

**Abstract** Maintenance of connective tissue integrity is fundamental to sustain function, requiring protein turnover to repair damaged tissue. However, connective tissue proteome dynamics remain largely undefined, as do differences in turnover rates of individual proteins in the collagen and glycoprotein phases of connective tissue extracellular matrix (ECM). Here, we investigate proteome dynamics in the collagen and glycoprotein phases of connective tissues by exploiting the spatially distinct fascicular (collagen-rich) and interfascicular (glycoprotein-rich) ECM phases of tendon. Using isotope labelling, mass spectrometry and bioinformatics, we calculate turnover rates of individual proteins within rat Achilles tendon and its ECM phases. Our results demonstrate complex proteome dynamics in tendon, with ~1000 fold differences in protein turnover rates, and overall faster protein turnover within the glycoprotein-rich interfascicular matrix compared to the collagen-rich fascicular matrix. These data provide insights into the complexity of proteome dynamics in tendon, likely required to maintain tissue homeostasis.

**\*For correspondence:**
cthorpe@rvc.ac.uk

[†]These authors contributed equally to this work

**Competing interests:** The authors declare that no competing interests exist.

## Introduction

Maintaining the structural and mechanical integrity of tissues in the musculoskeletal system, and other connective tissues, is fundamental to sustain tissue homeostasis and healthy function, requiring protein synthesis and degradation to repair and/or replace damaged tissue before damage accumulates and leads to injury (*Humphrey et al., 2014*). However, while the composition and structure of connective tissues is well defined (*Scott, 1983*), relatively little is known regarding proteome dynamics in connective tissues, particularly at the level of the individual constituent proteins.

Connective tissues consist of fibrous proteins (predominantly collagen) embedded in a glycoprotein-rich matrix (*Scott, 1983*), and variation in the organisation of both phases give rise to tissues with distinct structural and mechanical properties (*Culav et al., 1999*). Research in a variety of tissues including skin, tendon and cartilage indicates a relatively long half-life of collagens, with more rapid turnover of glycoproteins and other non-collagenous proteins (*Thorpe et al., 2010*; *Maroudas et al., 1998*; *Sivan et al., 2006*; *Sivan et al., 2008*; *Verzijl et al., 2000*). Indeed, several studies have reported negligible turnover of collagen, the major component of tendon and other connective tissues, within an individual's lifetime (*Thorpe et al., 2010*; *Heinemeier et al., 2013*;

**eLife digest** Muscles are anchored to bones through specialized tissues called tendons. Made of bundles of fibers (or fascicles) linked together by an 'interfascicular' matrix, healthy tendons are required for organisms to move properly. Yet, these structures are constantly exposed to damage: the interfascicular matrix, in particular, is highly susceptible to injury as it allows the fascicles to slide on each other.

One way to avoid damage could be for the body to continually replace proteins in tendons before they become too impaired. However, the way proteins are renewed in these structures is currently not well understood – indeed, it has long been assumed that almost no protein turnover occurs in tendons. In particular, it is unknown whether proteins in the interfascicular matrix have a higher turn over than those in the fascicles.

To investigate, Choi, Simpson et al. fed rats on water carrying a molecular label that becomes integrated into new proteins. Analysis of individual proteins from the rats' tendons showed great variation in protein turnover, with some replaced every few days and others only over several years. This suggests that protein turnover is actually an important part of tendon health. In particular, the results show that turnover is higher in the interfascicular matrix, where damage is expected to be more likely.

Protein turnover also plays a part in conditions such as cancer, heart disease and kidney disease. Using approaches like the one developed by Choi, Simpson et al. could help to understand how individual proteins are renewed in a range of diseases, and how to design new treatments.

*Heinemeier et al., 2016*). However other studies have measured relatively rapid collagen synthesis in tendon, both at basal levels and in response to exercise (*Miller et al., 2005*), and also identified soluble collagen with a much shorter half-life compared to the majority of collagen in human articular cartilage (*Hsueh et al., 2019*); the location(s) of these more labile proteins remain to be identified. When taken together, these contradictory findings indicate that proteome dynamics in connective tissues is complex, and that the glycoprotein-rich phase may be replenished more rapidly than the collagen-rich phase. However, the turnover rate of individual proteins within the different phases of the extracellular matrix (ECM), and the potential contribution of differential regulation of protein turnover to maintenance of tissue homeostasis remain undefined.

Tendon provides an ideal model in which to separately interrogate protein turnover in these ECM phases, as it consists of highly aligned, collagen-rich fascicular matrix (FM), interspersed by a less dense glycoprotein-rich phase, termed the interfascicular matrix (IFM, also referred to as the endotenon) (*Kastelic et al., 1978*). While the FM is predominantly composed of type I collagen, glycoproteins are also present in this region at low abundance (*Thorpe et al., 2016b*). Similarly, the IFM contains small amounts of a variety of collagens (*Södersten et al., 2013*; *Thorpe et al., 2016b*). Due to the highly aligned structure of tendon tissue, it is possible to separate FM and IFM for individual analysis using laser capture microdissection (*Thorpe et al., 2016b*; *Zamboulis et al., 2018*). Indeed, these, and other studies, have shown greater expression of markers of ECM degradation, as well as increased neo-peptide levels, a proxy measure of protein turnover, within the IFM (*Spiesz et al., 2015*; *Thorpe et al., 2015a*; *Thorpe et al., 2016b*), suggesting this region is more prone to micro-damage, likely due to the high shear environment that occurs due to interfascicular sliding (*Thorpe et al., 2012*; *Thorpe et al., 2015b*). Taken together, these findings suggest that there is greater turnover of ECM proteins localised to the IFM than within the FM to repair local microdamage and maintain tendon homeostasis.

Due to a lack of available methodology, it has not been possible until recently to study differential rate of turnover at the individual protein level. However, novel bioinformatics software developments, in combination with in vivo isotope labelling, and mass spectrometry technologies, now provide the capacity to determine the turnover rates of individual proteins within a sample (*Kim et al., 2012*; *Lam et al., 2014*; *Lau et al., 2016*). The aim of this study is therefore to apply this technique to tendon, firstly to establish the proteome-wide turnover rate of tendon, and secondly to combine this approach with laser capture microdissection to test the hypothesis that the proteome of the IFM is more dynamic than in the FM, with faster turnover of individual proteins. More rapid remodelling

of the IFM would provide a mechanism by which shear-induced microdamage to this region could be repaired, preventing damage accumulation and subsequent injury.

# Results

## $^2$H enrichment in serum water

In rats labelled with deuterium over a period of 127 days (*Figure 1a*), $^2$H enrichment of serum occurred rapidly and in a similar manner to that reported previously in rodents (*Kim et al., 2012*), reaching a plateau of 5.6% by day 4, and remained constant throughout the study (*Figure 1b*). The enrichment curve, which was empirically derived from the GC-MS measurements at the sampled time points, defines two parameters: deuterium enrichment rate ($k_p$) = 0.7913 and plateau ($p_{ss}$) = 0.0558; these values were used for subsequent kinetic curve fitting to calculate peptide turn-over rate constants ($k$).

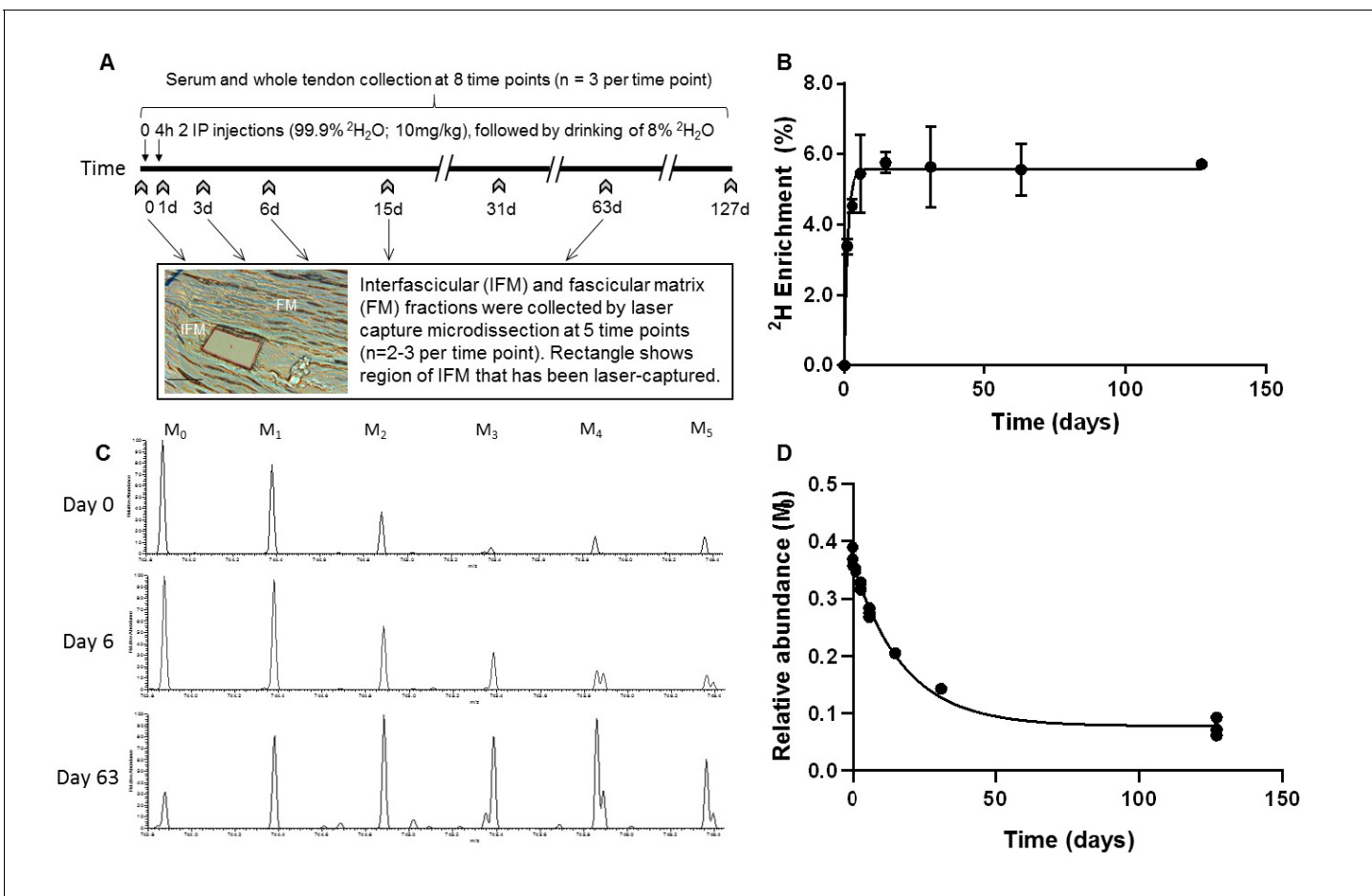

**Figure 1.** Metabolic labelling of rats using deuterium. (A) Schematic showing $^2$H$_2$O labelling of rats and details of sample collection, and interfascicular matrix isolation by laser capture microdissection (scale bar = 100 µm). (B) $^2$H enrichment in serum occurred rapidly, reaching a plateau of 5.6% by day 4 and remained constant throughout the study. Data are shown as mean ± SD and raw data provided in *Figure 1—source data 1*. (C) Example extracted ion chromatograms from tendon samples demonstrating increased abundance of higher mass isotopomer peaks over the course of the study (M$_1$ – M$_5$). (D) Example curve showing the relative abundance of the unlabelled monoisotopic peak (M$_0$) for the cartilage oligomeric matrix protein (COMP) peptide QMEQTYWQANPFR, calculated by ProTurn software and plotted as a function of time.

The online version of this article includes the following source data for figure 1:

**Source data 1.** Enrichment of serum water as measured by GC-MS.

## Protein identification

190 proteins with $\geq 2$ unique peptides were identified in whole tendon digests and protein interactions are shown in *Figure 2*. Of these proteins, 72 were classified as ECM or ECM-related proteins by MatrisomeDB (*Hynes and Naba, 2012*). In samples collected by laser capture microdissection of tendon cryosections, 266 proteins with $\geq 2$ unique peptides were identified in the IFM, 79 of which were ECM or ECM-related proteins (*Figure 3*). In the FM, 116 proteins were identified, of which 71 were ECM or ECM-related proteins (*Figure 4*). Protein interactions for each tendon component demonstrate a complex and highly interconnected proteome in tendon and its ECM phases.

## Protein turnover in whole tendon

Peptide turnover rates were calculated using ProTurn (v2.1.05; available at http://proturn.heartpro-teome.org; *Lam et al., 2014*; *Lau et al., 2016*; *Lau et al., 2018*; *Wang et al., 2014*), which automatically calculates turnover rate constants for all peptides that pass the selection criteria using non-steady state curve fitting (*Figure 1*). To assure data quality, we used a stringent cut-off, passing only peptides that are identified at 1% false discovery rate (FDR) and quantified at four or more time points. In total, 455 peptides, relating to 41 proteins, passed the ProTurn selection criteria and were used to calculate protein half-life in whole tendon samples.

The relative abundance of the unlabelled monoisotopic peak ($M_0$), plotted as a function of time for selected decorin and collagen peptides are shown in *Figure 5a&b*. Non-steady state curve fitting was performed by ProTurn to calculate turnover rate constants ($k$) for each peptide, and resultant fractional synthesis curves in *Figure 5c&d* demonstrate much faster turnover of decorin peptides compared to collagen types 1 and 3. $k$ values for all peptides identified, and corresponding protein half-lives are shown in *Figure 6*. As expected, the smallest $k$ values related to collagenous proteins, with corresponding half-lives of 330 to 1086 days. By contrast, the protein identified with the fastest turnover was the glycoprotein clusterin, with a half-life of 1.4 days. The half-life of proteoglycans ranged from 21 days for decorin to 72 days for lumican. With the exception of collagens, which all exhibited low turnover rates, there was no clear relationship between protein class and rate of turnover (*Figure 6c*).

## Protein turnover in tendon compartments

246 peptides, relating to 20 proteins, and 121 peptides, relating to 12 proteins, passed the ProTurn selection criteria in the FM and IFM respectively. 55 peptides were present both in the IFM and FM, and $k$ values were significantly greater for these peptides in the IFM compared to the FM (median: 0.018 vs. 0.010; p<0.0001), demonstrating an overall faster rate of protein turnover in the IFM. 39 peptides relating to collagen type I were identified in both the IFM and FM, with significantly greater $k$ values in the IFM (p<0.0001; *Figure 7a*). The turnover rate constants and resultant half-lives for proteins identified in each tendon phase are shown in *Figure 4b*. Turnover rate constants for Col1a1 and Col1a2 were significantly higher in the IFM compared to the FM, but there were no significant differences in rate constants for Col3a1 between tendon phases.

Due to ProTurn software identifying a low number of proteins in laser captured samples, particularly in the IFM, turnover rate constants in tendon phases were calculated manually for a number of proteins of interest (*Figure 8*). Rate constants of turnover of decorin peptides were significantly higher in the IFM compared to the FM (*Figure 8*). It was not possible to assess differences in turnover of other proteins due to a low number of peptides identified at sufficient time points to allow accurate curve fitting (*Figure 8c*).

## Discussion

This is the first study to investigate the proteome-wide turnover rate of tendon and its constituent ECM phases, demonstrating complex proteome dynamics in whole tendon. Protein turnover rate in tendon varied almost 1000-fold, with half-lives ranging from 1.5 days to over 1000 days. The results also support our hypothesis, demonstrating significantly faster turnover of proteins in the glycoprotein-rich IFM compared to the collagen-rich FM, both overall and when directly comparing individual proteins.

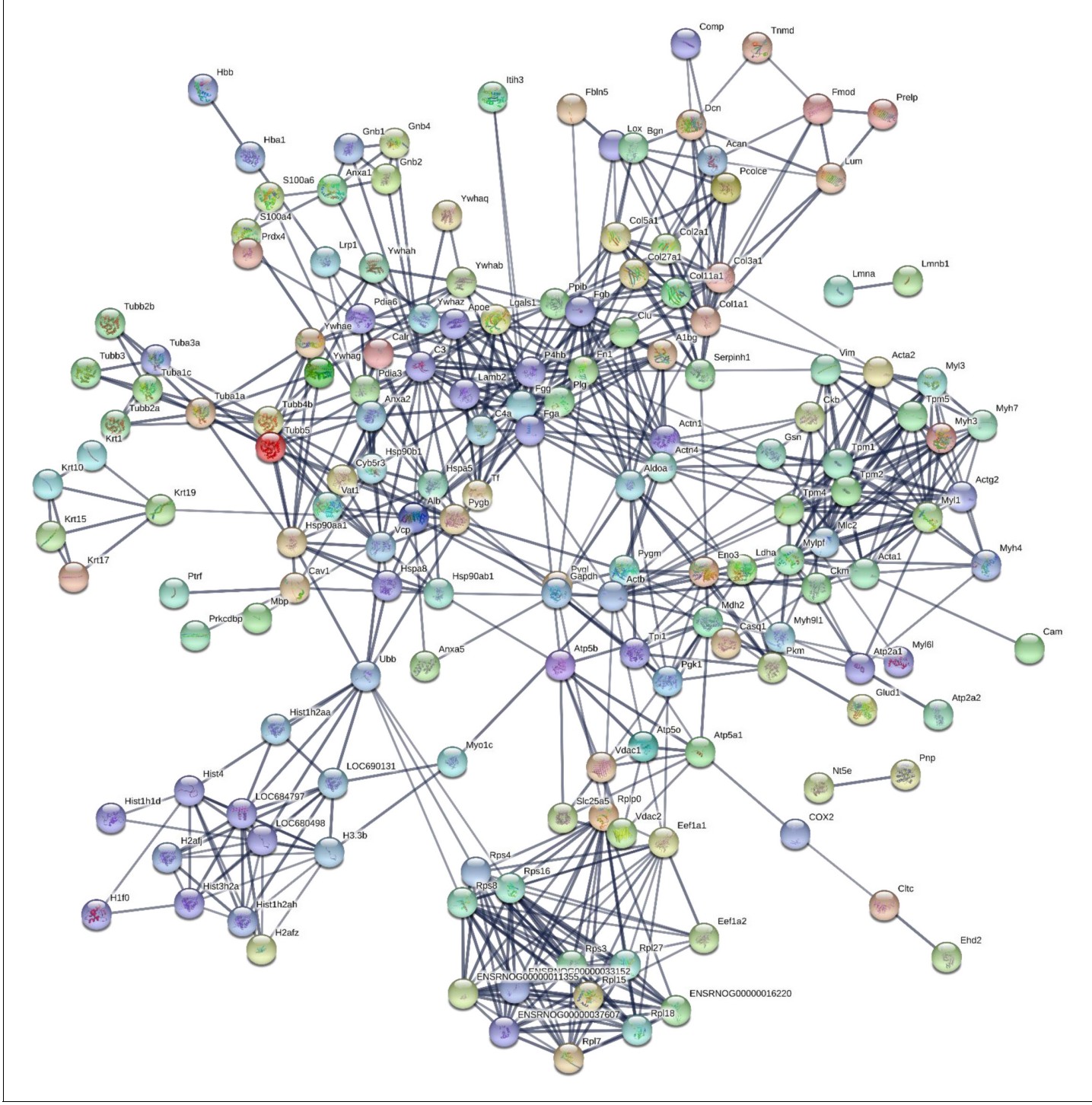

**Figure 2.** Protein-protein interaction map of proteins identified in whole tendon. Unconnected nodes were removed to provide clarity of the interactome. The total cluster was built with STRING (*Szklarczyk et al., 2019*) allowing for experimentally verified and predicted protein-protein interactions at high confidence levels (0.700). Line thickness indicates the strength of supporting data. Source data are provided in *Figure 2—source data 1*.

The online version of this article includes the following source data for figure 2:

**Source data 1.** Proteins identified in whole tendon samples used to create protein-protein interaction maps.

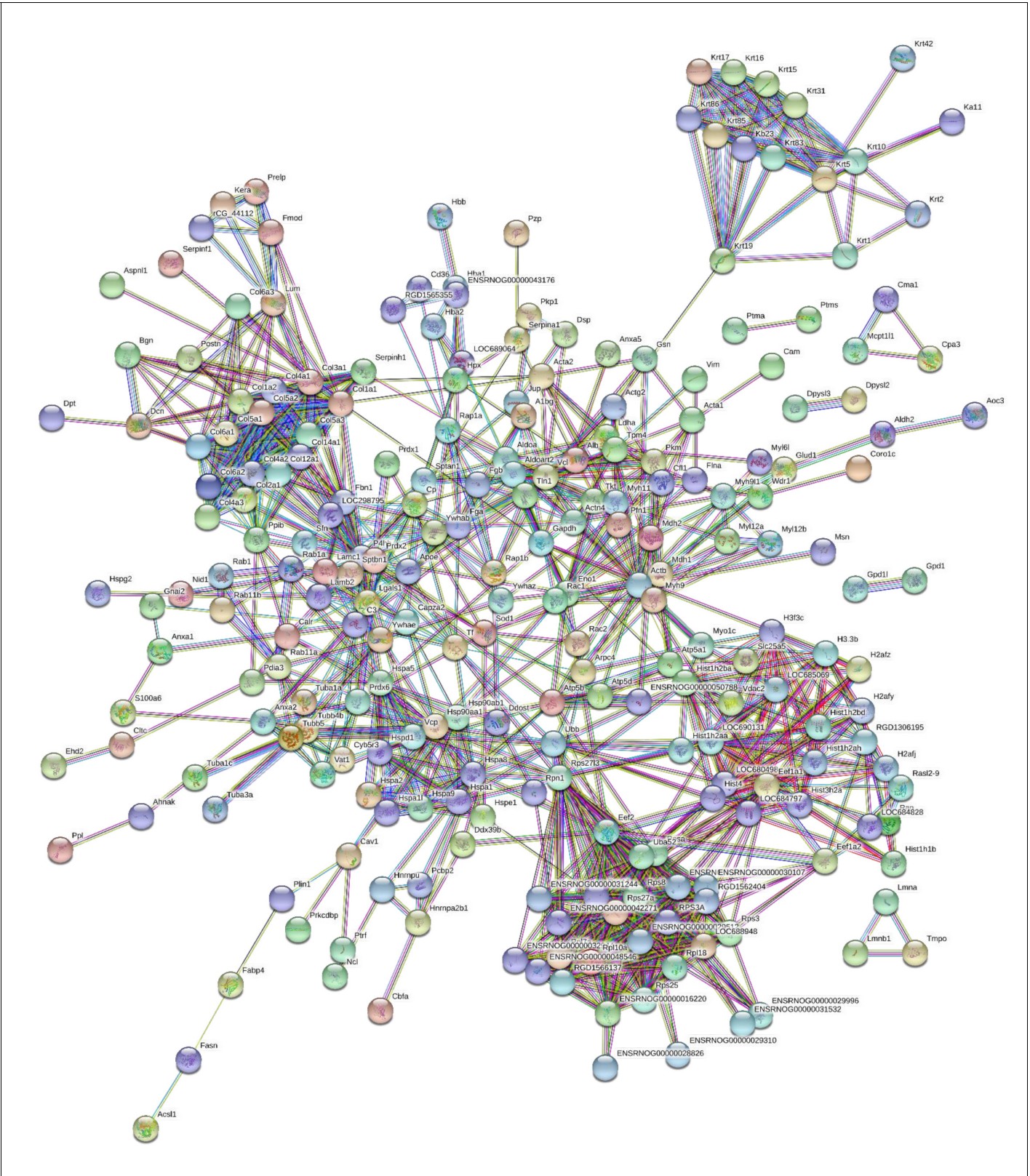

**Figure 3.** Protein-protein interaction map of proteins identified in the interfascicular matrix. Unconnected nodes were removed to provide clarity of the interactome. The total cluster was built with STRING (*Szklarczyk et al., 2019*) allowing for experimentally verified and predicted protein-protein interactions at high confidence levels (0.700). Line thickness indicates the strength of supporting data. Source data are provided in *Figure 3—source data 1*.

*Figure 3 continued on next page*

*Figure 3 continued*

The online version of this article includes the following source data for figure 3:

**Source data 1.** Proteins identified in the interfascicular matrix used to create protein-protein interaction maps.

In agreement with previous studies (*Thorpe et al., 2016b*), the IFM had a more complex proteome than the FM, with a greater number of proteins identified in this region. Indeed, more proteins were identified within the IFM than in whole tendon samples; this is likely due to the predominance of collagen in whole tendon samples which precludes detection of proteins present at lower abundance in these samples. As many of the proteins present in the IFM were identified in less than four samples, it was not possible to perform curve fitting to calculate the turnover rate for these proteins. Due to the small size of the rat Achilles tendon, it was not possible to laser-capture a larger volume of IFM, and pooling samples from each time point for LC-MS/MS analysis did not result in a greater number of protein hits than when each sample was analysed separately. In addition, it was necessary to calculate turnover rates of some proteins manually as automated identification of isotopomer peaks was unsuccessful, particularly in IFM samples. While turnover rates calculated manually cannot be compared directly to those calculated automatically by ProTurn, due to differences in curve fitting, it is possible to compare turnover rates of IFM and FM proteins, where both are calculated manually. Additionally, only tendons from female rats were used, such that any differences in turnover rate between sexes could not assessed. While no previous studies have directly compared tendon turnover rates in males and females, it has been demonstrated that there are very small differences in the transcriptome and proteome of Achilles tendons from male and female mice (*Sarver et al., 2017*). Despite these limitations, results reveal a diverse rate of protein turnover in both the FM and IFM, with values for individual proteins similar to those observed when the tendon matrix was analysed as a whole.

Collagenous proteins had the slowest turnover of all tendon matrix proteins identified, in the whole tendon, and in the FM and IFM, with a half-life from approximately 1.5 years in the IFM to 3 years in the FM for type I collagen. Collagen type III appeared to have a somewhat faster turnover rate than type I in both the FM and IFM, with no significant difference between phases. The IFM is known to be enriched in collagen type III, whereas the majority of collagen type I localises to the FM (*Södersten et al., 2013*; *Thorpe et al., 2016b*). While the collagen turnover rates measured are relatively slow, they contrast with previous studies which have reported that collagen is essentially inert in the adult rat patellar tendon (*Bechshøft et al., 2017*) and in tendons from larger animals (*Heinemeier et al., 2013*; *Thorpe et al., 2010*), as well as in cartilage and intervertebral disc (*Heinemeier et al., 2016*; *Sivan et al., 2008*). However, these previous studies have either analysed tissues as a whole, or separated the tissue into collagenous and non-collagenous fractions, rather than calculating turnover of individual proteins, such that they lack the specificity achieved by using deuterium labelling combined with mass spectrometry analysis. Other studies that have measured collagen synthesis in human tendon using stable isotope labelling and performing tendon biopsies, report contradictory findings, calculating a FSR of 0.04–0.06% hour$^{-1}$, which equates to a half-life ranging from 48 to 64 days (*Miller et al., 2005*; *Babraj et al., 2005*; *Smeets et al., 2019*). However, these were very short term studies, and it is unlikely that all newly synthesised protein would be incorporated into the matrix, such that FSR is over-estimated (*Heinemeier et al., 2013*). The results we present here help to explain these apparent contradictory findings, suggesting that the pool of more labile collagen identified previously in human tendon and cartilage (*Miller et al., 2005*; *Hsueh et al., 2019*) may be located in the glycoprotein-rich ECM phase rather than with the bulk of the fascicular collagen.

Another factor that is likely to influence tendon remodelling is the regulation of post-translational modifications. ECM proteins, particularly collagens, undergo a wide range of post-translational modifications during synthesis and assembly which occur both intra- and extra-cellularly, either through regulated pathways or spontaneously (*Yamauchi and Sricholpech, 2012*; *Avery and Bailey, 2005*). Post-translational modifications are essential for determining protein structure and function, and are also likely to influence susceptibility to degradation by limiting enzyme access to cleavage sites (*Jaisson et al., 2007*). Indeed, a previous study has shown that, with ageing, partially degraded collagen accumulates within the equine superficial digital flexor tendon (*Thorpe et al., 2010*). This is

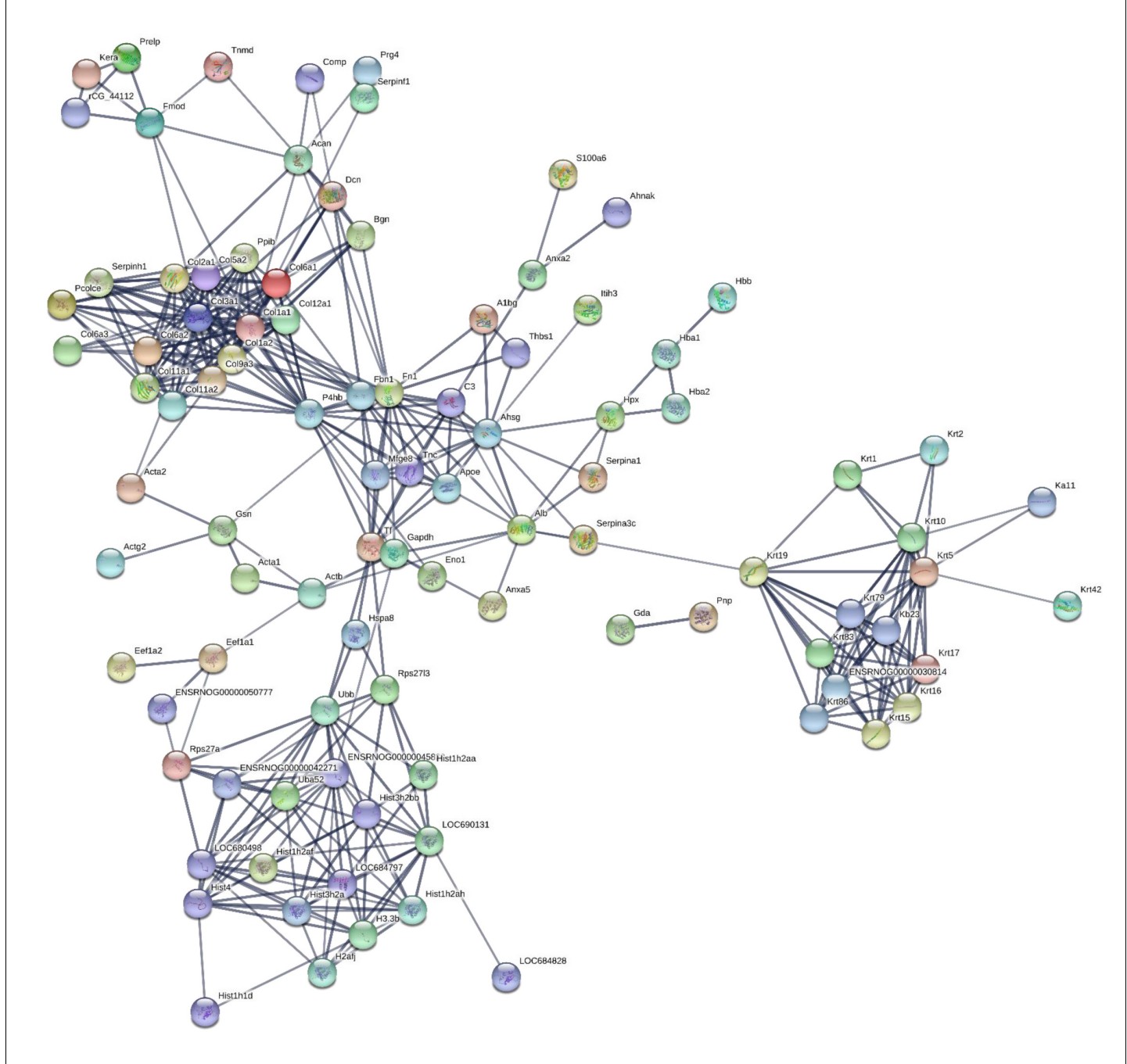

**Figure 4.** Protein-protein interaction map of proteins identified in the fascicular matrix. Unconnected nodes were removed to provide clarity of the interactome. The total cluster was built with STRING (*Szklarczyk et al., 2019*) allowing for experimentally verified and predicted protein-protein interactions at high confidence levels (0.700). Line thickness indicates the strength of supporting data. Source data are provided in *Figure 4—source data 1*.

The online version of this article includes the following source data for figure 4:

**Source data 1.** Proteins identified in the fascicular matrix used to create protein-protein interaction maps.

accompanied by increased advanced glycation end products which may render the ECM more resistant to degradation and subsequent remodelling, reducing the ability to repair damage within the tendon (*Thorpe et al., 2010*). However, the effects of alterations in post-translations modifications on rates of protein turnover remain to be determined.

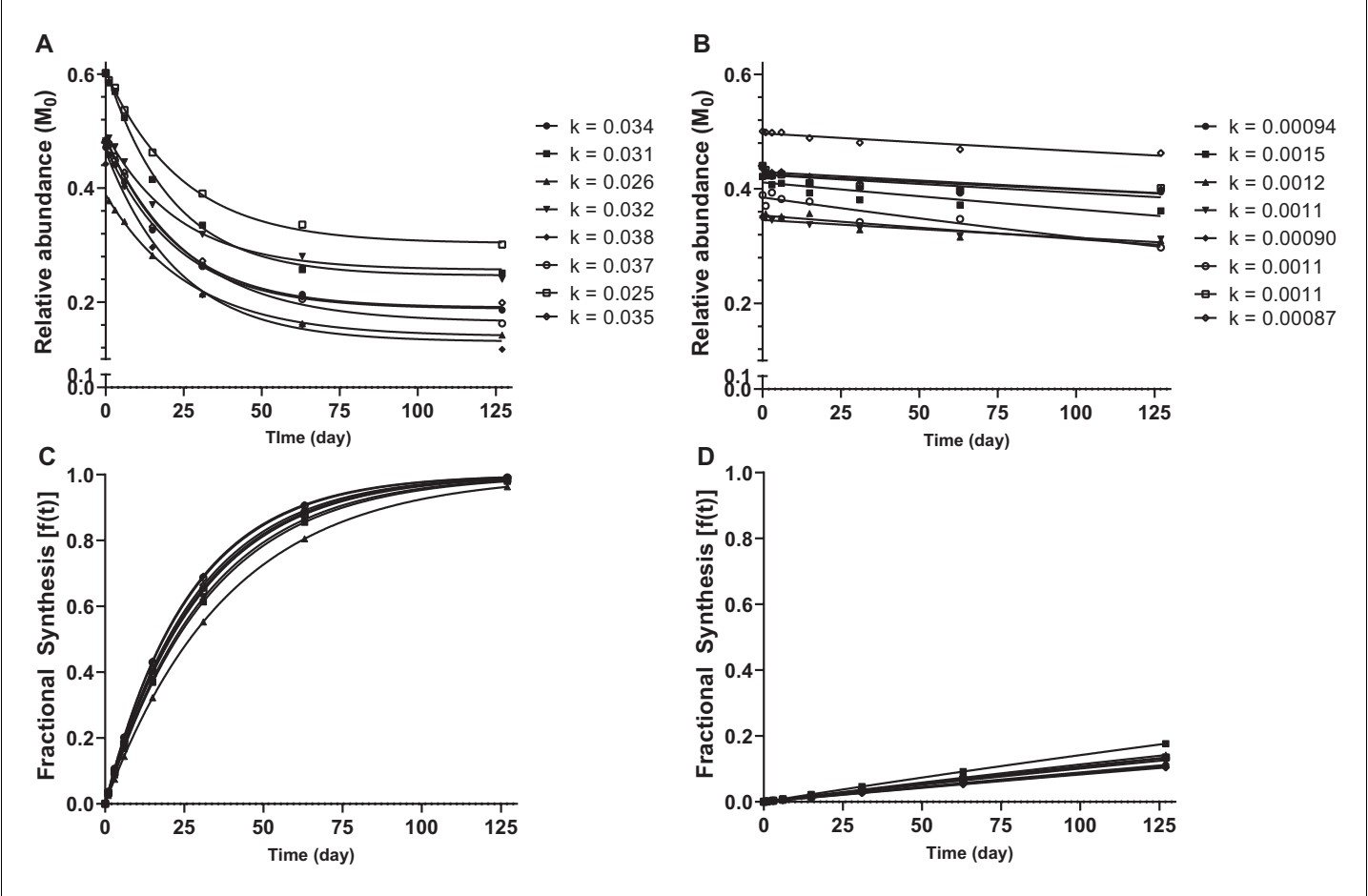

**Figure 5.** Calculation of protein turnover rate from mass isotopomer distribution over time. Relative abundance of $M_0$ in selected decorin (**A**) and collagen type 1, alpha one chain (**B**) peptides, and resulting *k* values calculated by non-steady state curve fitting using ProTurn software. Fractional synthesis rates (FSR), calculated from peptide *k* values, demonstrate more rapid turnover of decorin (**C**) compared to collagen type 1, alpha one chain (**D**). Source data, generated by ProTurn are available in *Source data 1*.

In addition, it is likely that overall protein turnover rate is more rapid in small compared to large animals. While few studies have directly compared protein turnover kinetics between species, a recent study of protein turnover in dermal fibroblasts from different rodent species demonstrated a negative correlation between median *k* values and lifespan (*Swovick et al., 2018*). In support of this, the half-life of serum albumin is approximately 10 fold greater in the human compared to the rat (*Chaudhury et al., 2003*; *Jeffay, 1960*). Taken together, it can be presumed that turnover of tendon ECM proteins is also likely slower in larger animals than those we report here for the rat, but the relative differences observed between individual proteins and ECM phases are likely to persist in humans and other large animals.

It is also important to recognise that specific proteins that are not core components of the tendon ECM will not be metabolised within the tendon, but instead will be synthesised elsewhere and transported to the tendon via the circulation. Specifically, serum albumin and serotransferrin are predominantly synthesised and secreted by the liver (*Tavill, 1972*), whereas haemoglobin is synthesised in the bone marrow and its turnover provides a measure of erythrocyte lifespan (*Koury, 2016*). In addition, anionic trypsin-1, which had a particularly long half-life of nearly 2000 days in the FM, is synthesised in the pancreas. The long half-life measured for this protein is rather surprising, as generally enzymes have a much shorter half-life than ECM proteins (*Lake-Bakaar et al., 1980*) Proteases have a role in tissue repair (*Shah and Mital, 2018*); it is possible that there is a pool of trypsin within tendon that remains inactive until required for repair, which would explain the long half-life measured.

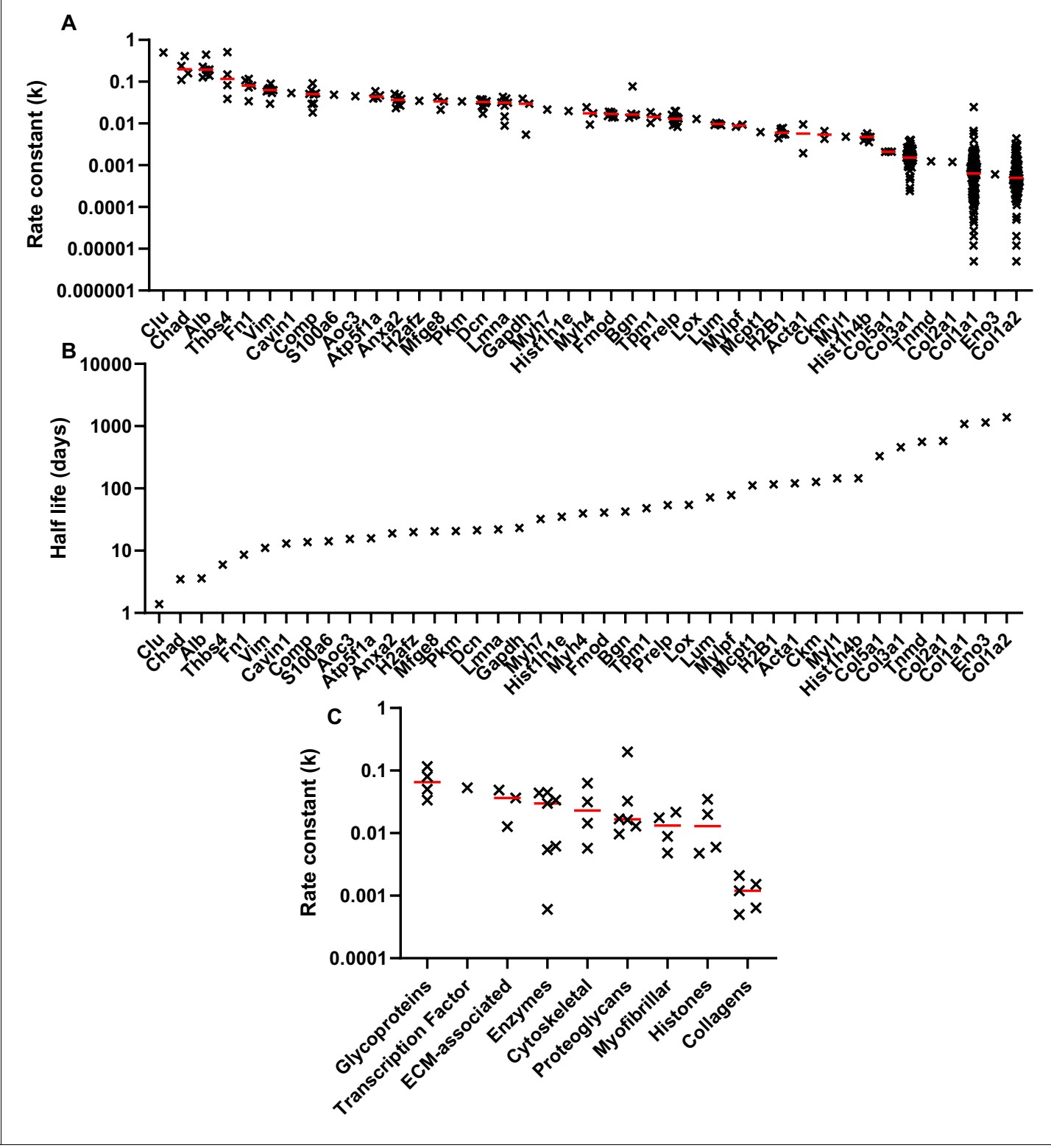

**Figure 6.** Peptide turnover rate constants and corresponding protein half-lives in whole tendon. (A) The peptide rate constants (k) for individual proteins are plotted in descending order on a logarithmic scale, with the median value represented by a red line. (B) The median *k* values for each protein were used to calculate protein half-life, assuming a first order reaction. (C) Protein turnover rates plotted against MatrisomeDB and PANTHER categories, with the median value represented by a red line. Due to space constraints, gene, rather than protein, names are displayed in parts *A* and *B*. Source data, generated by ProTurn are available in *Source data 1*.

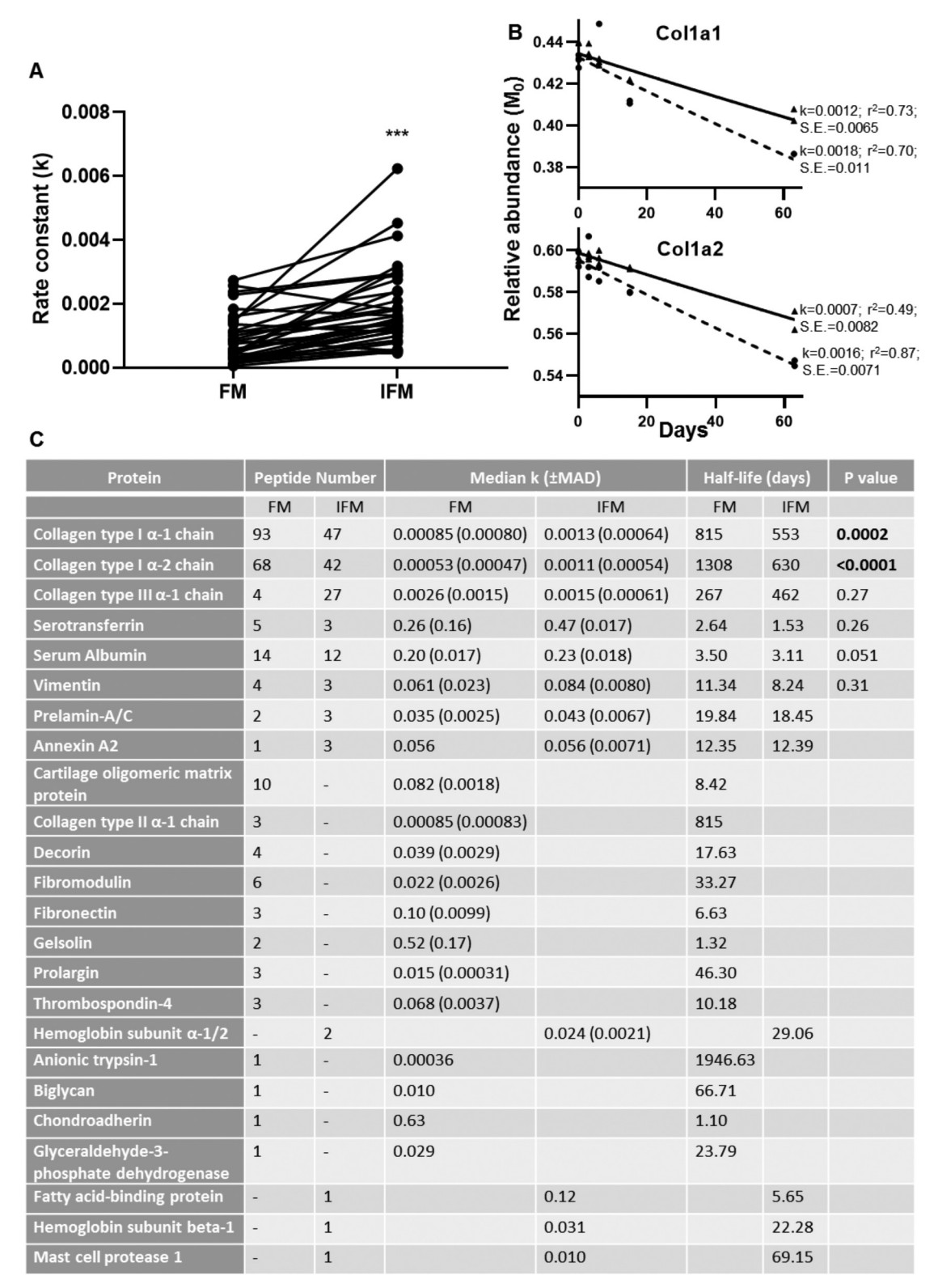

| Protein | Peptide Number | | Median k (±MAD) | | Half-life (days) | | P value |
|---|---|---|---|---|---|---|---|
| | FM | IFM | FM | IFM | FM | IFM | |
| Collagen type I α-1 chain | 93 | 47 | 0.00085 (0.00080) | 0.0013 (0.00064) | 815 | 553 | **0.0002** |
| Collagen type I α-2 chain | 68 | 42 | 0.00053 (0.00047) | 0.0011 (0.00054) | 1308 | 630 | **<0.0001** |
| Collagen type III α-1 chain | 4 | 27 | 0.0026 (0.0015) | 0.0015 (0.00061) | 267 | 462 | 0.27 |
| Serotransferrin | 5 | 3 | 0.26 (0.16) | 0.47 (0.017) | 2.64 | 1.53 | 0.26 |
| Serum Albumin | 14 | 12 | 0.20 (0.017) | 0.23 (0.018) | 3.50 | 3.11 | 0.051 |
| Vimentin | 4 | 3 | 0.061 (0.023) | 0.084 (0.0080) | 11.34 | 8.24 | 0.31 |
| Prelamin-A/C | 2 | 3 | 0.035 (0.0025) | 0.043 (0.0067) | 19.84 | 18.45 | |
| Annexin A2 | 1 | 3 | 0.056 | 0.056 (0.0071) | 12.35 | 12.39 | |
| Cartilage oligomeric matrix protein | 10 | - | 0.082 (0.0018) | | 8.42 | | |
| Collagen type II α-1 chain | 3 | - | 0.00085 (0.00083) | | 815 | | |
| Decorin | 4 | - | 0.039 (0.0029) | | 17.63 | | |
| Fibromodulin | 6 | - | 0.022 (0.0026) | | 33.27 | | |
| Fibronectin | 3 | - | 0.10 (0.0099) | | 6.63 | | |
| Gelsolin | 2 | - | 0.52 (0.17) | | 1.32 | | |
| Prolargin | 3 | - | 0.015 (0.00031) | | 46.30 | | |
| Thrombospondin-4 | 3 | - | 0.068 (0.0037) | | 10.18 | | |
| Hemoglobin subunit α-1/2 | - | 2 | | 0.024 (0.0021) | | 29.06 | |
| Anionic trypsin-1 | 1 | - | 0.00036 | | 1946.63 | | |
| Biglycan | 1 | - | 0.010 | | 66.71 | | |
| Chondroadherin | 1 | - | 0.63 | | 1.10 | | |
| Glyceraldehyde-3-phosphate dehydrogenase | 1 | - | 0.029 | | 23.79 | | |
| Fatty acid-binding protein | - | 1 | | 0.12 | | 5.65 | |
| Hemoglobin subunit beta-1 | - | 1 | | 0.031 | | 22.28 | |
| Mast cell protease 1 | - | 1 | | 0.010 | | 69.15 | |

**Figure 7.** Peptide turnover rate constants and corresponding protein half-lives in tendon phases. (A) Rate constants (k) for collagen type I peptides identified in both tendon phases were significantly greater in the IFM compared to the FM (n = 39). (B) Median peptide decay curves for Col1a1 and Col1a2 in the FM (▲; solid line) and IFM (•; dashed line), showing goodness of fit ($r^2$) and standard error (S.E.). (C) Turnover rate constants (k) and

*Figure 7 continued on next page*

*Figure 7 continued*

corresponding half-lives for proteins identified in each phase. *** indicates p<0.0001. Source data, generated by ProTurn are available in *Figure 7—source data 1* for the FM and *Figure 7—source data 2* for the IFM.
The online version of this article includes the following source data for figure 7:

**Source data 1.** ProTurn output for FM.
**Source data 2.** ProTurn output for IFM.

The measured rate constants of protein turnover will also be dependent on the protein extraction methods used. We chose GuHCl extraction for whole tendon as this has previously been shown to result in best proteome coverage for tendon (*Ashraf Kharaz et al., 2017*). It was not possible to use the same extraction method for laser-captured samples due to the small amount of starting material, therefore we used a method previously optimised for this sample type (*Thorpe et al., 2016b*).

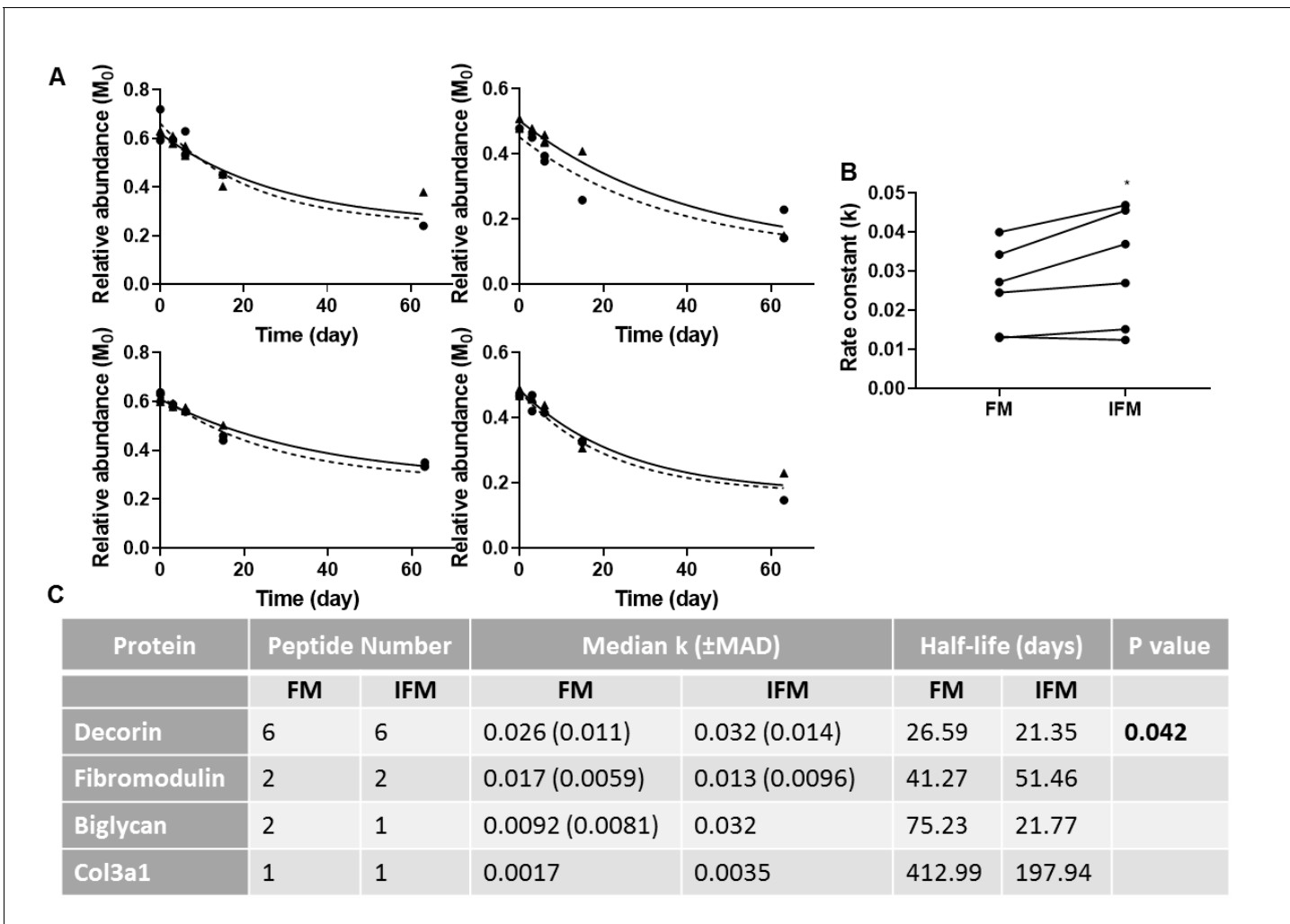

| Protein | Peptide Number | | Median k (±MAD) | | Half-life (days) | | P value |
|---|---|---|---|---|---|---|---|
| | FM | IFM | FM | IFM | FM | IFM | |
| Decorin | 6 | 6 | 0.026 (0.011) | 0.032 (0.014) | 26.59 | 21.35 | **0.042** |
| Fibromodulin | 2 | 2 | 0.017 (0.0059) | 0.013 (0.0096) | 41.27 | 51.46 | |
| Biglycan | 2 | 1 | 0.0092 (0.0081) | 0.032 | 75.23 | 21.77 | |
| Col3a1 | 1 | 1 | 0.0017 | 0.0035 | 412.99 | 197.94 | |

**Figure 8.** Manual calculation of turnover rate of selected proteins. (**A**) Relative abundance of $M_0$ as a function of time, and resultant non-linear curve fitting for peptides corresponding to decorin identified in the FM (▲; solid line) and IFM (•; dashed line). (**B**) Resultant k values for decorin peptides were significantly greater in the IFM than in the FM (p=0.042). (**C**) Manually calculated turnover rate constants (k) and corresponding half-lives for tendon proteins of interest. Source data are available in *Figure 8—source data 1*.
The online version of this article includes the following source data for figure 8:

**Source data 1.** GraphPad Prism output showing the manually calculated relative abundance of $M_0$ at different time points for decorin, fibromodulin, biglycan and Col3a1 peptides, and resultant K values.

Despite differences in extraction techniques used, protein half-lives were similar in the whole tendon and the fascicular matrix, suggesting minimal effects from using different extraction techniques. Indeed, previous studies, in which the rat Achilles was acid-solubilised post deuterium labelling and FSR measured by GC-MS, reported a FSR of 0.66 % day$^{-1}$ (*Katsma et al., 2017*) which equates to a half-life of 105 days, supporting the results we present here, in which average protein half-life in whole tendon is 131 days.

As well as providing the first data on individual protein turnover rates in tendon, this work builds on our understanding of the role of the IFM in tendon function. Previous work has shown that the IFM has specialised composition and function, facilitating sliding between tendon fascicles to provide extensibility to the tendon (*Thorpe et al., 2012*; *Thorpe et al., 2015b*; *Thorpe et al., 2016b*; *Thorpe et al., 2016c*; *Thorpe et al., 2016a*). Cyclic sliding and recoil increases the risk of microdamage to this region (*Spiesz et al., 2015*), and therefore greater turnover of IFM proteins may be a protective mechanism to repair this damage. We have previously demonstrated that elastin and lubricin are localised to the IFM and likely contribute to its mechanical function (*Godinho et al., 2017*; *Thorpe et al., 2016a*). However, these proteins are difficult to identify using LC-MS/MS (*Thorpe et al., 2016b*) so we were unable to assess their rate of turnover.

We extend previous observations in tendon and other musculoskeletal tissues (*Thorpe et al., 2010*; *Sivan et al., 2006*; *Maroudas et al., 1998*; *Hsueh et al., 2019*), measuring considerably faster turnover of proteoglycans compared to collagens in tendons, FM and IFM. Interestingly, there were variations in rates of small leucine-rich proteoglycan turnover between tendon phases. Significantly faster turnover of decorin was observed in the IFM compared to the FM, whereas the results suggest that faster fibromodulin turnover may occur within the FM. Indeed, previous studies have shown differential abundance of specific proteoglycans in the IFM and FM, with enrichment of fibromodulin in the FM, but no significant differences in amounts of decorin, biglycan or lumican between tendon phases, as determined by proteomic analysis and immunohistochemistry (*Thorpe et al., 2016b*; *Thorpe et al., 2016a*). While use of mouse knock out models have established the roles of a variety of proteins during tendon development, with collagen type V, decorin, biglycan, fibromodulin, lumican and thrombospondins all serving roles in modulating collagen fibrillogenesis (*Connizzo et al., 2013*; *Delgado Caceres et al., 2018*), the contributions of these proteins to the function of mature tendon are less well characterised. Recent work using decorin and biglycan inducible knock-out mice demonstrated alterations in collagen fibril diameter and decreased tendon mechanical properties in the adult, indicating an important role for these proteoglycans in FM and subsequently, tendon homeostasis (*Robinson et al., 2017*). It is likely that proteoglycans also contribute to homeostasis within the IFM, but the roles of specific proteins in the IFM during development and in the adult remain to be defined.

Our results show that proteome dynamics in tendon is highly complex, with differences in protein-selective turnover rates observed in distinct IFM and FM ECM phases. Regulation of protein turnover is multifactorial, involving pathways controlling protein synthesis, folding, transport, assembly, disassembly, and degradation (*Powers et al., 2009*). Recent studies have identified the important role of transforming growth factor-β (TGF-β) in regulating ECM remodelling in tendon and other tissues (*Gumucio et al., 2015*; *Rys et al., 2016*). Indeed, it has been shown that TGF-β localises to the IFM in developing and adult tendon (*Kuo et al., 2008*; *Russo et al., 2015*), which is consistent with the notion that increased TGF-β signalling could have a role in this faster protein turnover identified in the IFM.

## Conclusions

This is the first study to determine the turnover rate of individual proteins in different connective tissue phases, demonstrating complex proteome dynamics in tendon, with an overall faster turnover of proteins in the glycoprotein-rich IFM phase. Greater capacity for turnover of this phase of the tendon matrix may be indicative of a mechanism that reduces the accumulation of damage caused by the high shear environment within this region, although this remains to be directly determined. The techniques used here provide a powerful approach to interrogate alterations in connective tissue protein turnover with ageing and/or disease which are likely to influence injury risk.

# Materials and methods

**Key resources table**

| Reagent type (species) or resource | Designation | Source or reference | Identifiers | Additional information |
|---|---|---|---|---|
| Strain, strain background (*Rattus Norvegicus, Female*) | Wistar | Charles River | RRID:RGD_13508588 | Female |
| Commercial assay or kit | Pierce protein assay | ThermoFisher | 22660 | |
| Chemical compound, drug | $[^2H]_2O$ | CK isotopes LtD | DLM-2259 | |
| Chemical compound, drug | Acetonitrile with 0.1% formic acid LCMS grade | Fisher Scientific | 10723857 | |
| Chemical compound, drug | Ammonium bicarbonate | Sigma | 09830 | |
| Chemical compound, drug | Chondroitinase ABC | Sigma | C3667 | |
| Chemical compound, drug | Dithiothreitol | Melford laboratories | MB1015 | |
| Chemical compound, drug | Formic acid (0.1% v/v) LCMS grade | Fisher Scientific | 10188164 | |
| Chemical compound, drug | Guanidine hydrocholoride | Sigma | G3272 | |
| Chemical compound, drug | Iodoacetamide | Sigma Aldrich | I1149 | |
| Chemical compound, drug | RapiGest SF | Waters | 186001861 | |
| Chemical compound, drug | Trifluoroacetic acid Optima | Fisher Scientific | 10723857 | |
| Chemical compound, drug | Trifluoroacetic acid (0.1% v/v) LCMS grade | Fisher Scientific | 10516625 | |
| Chemical compound, drug | Trypsin Gold MS grade | Promega | V5280 | |
| Software, algorithm | Peaks Studio v8.5 | Bioinformatics Solutions | www.bioinfor.com/peaks-studio | |
| Software, algorithm | STRING v11.0 | PMID:30476243 | string-db.org | |
| Software, algorithm | MatrisomeDB | PMID:21937732 | matrisomeproject.mit.edu | |
| Software, algorithm | PANTHER | PMID:23868073 | www.pantherdb.org | |
| Software, algorithm | ProLuCID | PMID:26171723 | fields.scripps.edu/yates/wp/?page_id = 821 | |
| Software, algorithm | ProteoWizard v3 | PMID:28188540 | proteowizard.sourceforge.net/index.html | |
| Software, algorithm | ProTurn | PMID:24614109 | proturn.heartproteome.org | |
| Software, algorithm | Prism v8.2 | GraphPad | www.graphpad.com | |
| Other | Filter units, Vivacon 500 10000 MWCO | Sartorius | VN01H02 | |

## Animals

- Female Wistar rats (n = 24, 12 week old, weight: 270 ± 18 g, range: 228–299 g, specific pathogen free, Charles River Company, UK) were randomly housed in polypropylene cages in groups of 3,

subjected to 12 hr light/dark cycle with room temperature at $21 \pm 2°C$ and fed ad libitum with a maintenance diet (Special Diet Services, Witham UK). All procedures complied with the Animals (Scientific Procedures) Act 1986, were approved by the local ethics committee at the Royal Veterinary College, were performed under project licence PB78F43EE and are reported according to the ARRIVE guidelines (*Kilkenny et al., 2010*).

## In vivo isotope labelling

All rats, with the exception of the controls (n = 3) received two intraperitoneal injections of 99% $[^2H]_2O$ (10 ml/kg; CK isotopes Ltd) spaced 4 hr apart, and were then allowed free access to 8% $[^2H]_2O$ (v/v) in drinking water for the duration of the study to maintain steady-state labelling (*Kim et al., 2012*). All rats were acclimatized for 1 week prior to commencement of the study, and monitored daily and weighed weekly throughout the study; no adverse effects were observed. Rats in the control group were sacrificed on day 0. Rats in the isotope labelled groups were sacrificed on day 1, 3, 6, 15, 31, 63 and 127 (n = 3 per time point). Rats were culled at the same time of day (10 am) at each time point, to minimise any effect from diurnal variations. Blood was collected from each rat immediately post-mortem, and allowed to clot at room temperature. Blood samples were centrifuged at 1500 g for 10 min, and serum collected and stored at $-20°C$ prior to analysis. Both Achilles tendons were harvested within 2 hr of death. The left Achilles was snap frozen in n-hexane cooled on dry ice for proteomic analysis of the whole tissue, and the right Achilles was embedded in optimal cutting temperature compound and snap frozen in n-hexane cooled on dry ice for isolation of fascicular and interfascicular matrices (*Figure 1a*).

## GC-MS analysis of serum water

$^2H$ labelling in serum was measured via gas chromatography-mass spectrometry (GC-MS) after exchange with acetone and extraction with chloroform as described previously (*McCabe et al., 2006*). 30 μl extract was analysed using GC-MS (Waters GCT mass spectrometer, Agilent J and W DB-17MS column (30m $\times$ 0.25 mm x 0.25 μm)). The carrier gas was helium (flow rate: 0.8 ml/min). The column temperature gradient was as follows: 60°C initial, 20 °C/min increase to 100°C, 1 min hold, then 50 °C/min increase to 220°C. The injection volume was 1 μl and injector temperature was 220°C. The mass spectrometer operated in positive ion electron ionisation mode, the source temperature was 180°C and the range scanned was m/z 40–600 (scan time: 0.9 s). Mass spectral intensities for m/z values 58 and 59 were produced by combining the mass spectra in the acetone peak to create an averaged spectrum (Waters MassLynx). Serum $^2H$ enrichment was calculated by comparison to a standard curve, and first order curve fitting (GraphPad Prism 8) used to calculate the rate constant ($k_p$) and plateau ($p_{ss}$) of deuterium enrichment (*Figure 1b*).

## Protein extraction from whole tendon

Left Achilles tendons were thawed, and their surface scraped with a scalpel, followed by PBS washes, to remove the epitenon and any residual non-tendinous tissue. Tendons were finely chopped, flash frozen in liquid nitrogen and pulverised in a dismembrator (Sartorius, mikro-dismembrator U, 1800rpm, 2 mins.). Proteins were extracted using previously optimised methodologies (*Ashraf Kharaz et al., 2017*). Briefly, following deglycosylation in chondroitinase ABC, proteins were extracted in guanidine hydrocholoride (GuHCl) as described previously (*Kharaz et al., 2016*). Protein content was determined using a Pierce protein assay according to the manufacturer's instructions. Samples were stored at $-80°C$ prior to preparation for LC-MS/MS by centrifugal filtration. Filter units (Vivacon 500; 10 000 MWCO) were rinsed with 1%(v/v) formic acid and a balance volume of buffer (4M GuHCl/50 mM ammonium bicarbonate (ambic)) was added. A volume equivalent to 50 μg protein was added to each filter and the samples vortexed gently. Filters were centrifuged (15 min, 12500 rpm, 20°C), washed with GuHCl buffer and centrifuged again. Proteins were reduced by DTT incubation (100 μl of 8 mM in 4M GuHCl, 15 min, 56°C) and then centrifuged (10 min, 12500 rpm). DTT was removed by washing twice with buffer as described above. Proteins were alkylated with 100 μl 50 mM iodoacetamide in 4M GuHCl solution, vortexed and incubated in the dark (20 min., room temperature). Iodoacetamide was removed by washing twice with GuHCl buffer as described above. Buffer was exchanged to ambic by 3 washes with 50 mM ambic, centrifuging after each wash. 1 μg trypsin in 50 mM ambic (40 μl) was added and proteins digested overnight at 37°C with

mixing. Flow through was collected after centrifugation and addition of 40 µl 50 mM ambic. The flow though was combined and acidified using trifluoroacetic acid (10%(v/v)), and diluted 20-fold in 0.1%(v/v) TFA/3%(v/v) acetonitrile for LC-MS/MS analysis.

## Isolation and extraction of fascicular and interfascicular matrix proteins

 Longitudinal cryosections (15 µm) from the right Achilles tendons (five time points, n = 2–3/time point) were adhered to membrane slides (PEN, Leica) and stored at −80˚C. Sections were prepared for laser capture as described previously (*Thorpe et al., 2016b*) and approximately 1 mm$^2$ of FM and IFM from each sample were collected into 50 µl molecular biology grade H$_2$O (Leica LMD7000; *Figure 1a*). Samples were immediately frozen and stored at −80˚C. Prior to digestion, samples were centrifuged immediately after removal from −80˚C storage. Volume was adjusted to 80 µl by adding 50 mM ambic, and Rapigest SF (1%(v/v) in 25 mM ambic, Waters, UK) was added to a final volume of 0.1%(v/v). Samples were mixed (room temperature, 30 min) then incubated at 60˚C (1 hr, 450 rpm). Samples were centrifuged (17200 g, 10 min), vortexed, then incubated at 80˚C (10 min). Proteins were reduced by incubating with DTT (5 µL, 60 mM in 25 mM ambic, 10 min, 60˚C) then alkylated by adding 5 µl 178 mM iodoacetamide in 25 mM ambic (30 min. incubation, room temperature, in dark). Trypsin (Promega Gold sequencing grade) was diluted in 25 mM ambic and added at an enzyme to protein ratio of 1:50 (based on estimated protein amount from tissue volume collected). Digests were incubated overnight at 37˚C with an enzyme top-up after 3.5 hr. Rapigest was hydrolysed with TFA (0.5 µl, 37˚C, 45 min.). Digests were centrifuged (17200 g, 30 min.), aspirated into low-bind tubes and de-salted on stage-tips as previously described (*Thorpe et al., 2016b*).

## LC-MS/MS analyses

Data-dependent LC-MS/MS analyses were conducted on a QExactive quadrupole-Orbitrap mass spectrometer (*Williamson et al., 2016*; *Michalski et al., 2011*) coupled to a Dionex Ultimate 3000 RSLC nano-liquid chromatograph (Thermo Fisher, UK). Digest was loaded onto a trapping column (Acclaim PepMap 100 C18, 75 µm x 2 cm, 3 µm packing material, 100 Å) using a loading buffer of 0.1%(v/v) TFA, 2%(v/v) acetonitrile in water for 7 min (flow rate: 9 µl/min). The trapping column was set in-line with an analytical column (EASY-Spray PepMap RSLC C18, 75 µm x 50 cm, 2 µm packing material, 100 Å) and peptides eluted using a linear gradient of 96.2% A (0.1%(v/v) formic acid):3.8% B (0.1%(v/v) formic acid in water:acetonitrile [80:20](v/v)) to 50% A:50% B over 30 min (flow rate: 300 nl/min), followed by 1% A:99% B for 5 min and re-equilibration of the column to starting conditions. The column was maintained at 40˚C, and the effluent introduced directly into the integrated nano-electrospray ionisation source operating in positive ion mode. The mass spectrometer was operated in DDA mode with survey scans between *m/z* 300–2000 acquired at a mass resolution of 70,000 (FWHM) at *m/z* 200. The maximum injection time was 250 ms, and the automatic gain control was set to 1e6. The 10 most intense precursor ions with charges states of between 2+ and 4+ were selected for MS/MS with an isolation window of 2 *m/z* units. The maximum injection time was 100 ms, and the automatic gain control was set to 1e5. Peptides were fragmented by higher-energy collisional dissociation using a normalised collision energy of 30%. Dynamic exclusion of *m/z* values to prevent repeated fragmentation of the same peptide was used (exclusion time: 20 s).

## Peptide and protein identification

Proteins were identified from RAW data files, with trypsin specified as the protease, with one missed cleavage allowed and with fixed modifications of carbamidomethyl cysteine and variable modifications of oxidation of methionine and proline (Peaks Studio v8.5, Bioinformatics Solutions, Waterloo, Canada). Searches were performed against the UniProt *Rattus Norvegicus* database (www.uniprot.org/proteomes), with an FDR of 1%,≥2 unique peptides per protein and a confidence score >20. Protein network analysis was performed using the Search Tool for Retrieval of Interacting Genes/Proteins (STRING), v11.0 (*Szklarczyk et al., 2019*), and proteins were further classified using MatrisomeDB (*Hynes and Naba, 2012*) and the Protein ANalysis THrough Evolutionary Relationships (PANTHER) Classification System (*Mi et al., 2013*). The proteomic data have been deposited to the ProteomeXchange Consortium via the PRIDE partner repository (*Perez-Riverol et al., 2019*) with the data set identifier PXD015928 and 10.6019/PXD015928.

## Calculation of protein turnover rates

Peptides at each time point were identified using ProLuCID (*Xu et al., 2015*) searching against a reverse-decoyed protein sequence database (UniProt *Rattus Norvigicus*, reviewed, accessed 23/04/2018). Fixed modifications of carbamidomethyl cysteine and ≤3 variable modifications of oxidation of methionine and proline were allowed. Tryptic, semi-tryptic, and non-tryptic peptides within a 15-ppm mass window surrounding the candidate precursor mass were searched. Protein identification was performed using DTASelect (v.2.1.4) (*Cociorva et al., 2006*), with ≤1% global peptide FDR and ≥2 unique peptides per protein. Protein turnover rates and resulting half-lives were calculated using custom software (ProTurn v2.1.05; available at: http://proturn.heartproteome.org) as previously described and validated (*Lam et al., 2014*; *Lau et al., 2016*; *Lau et al., 2018*; *Wang et al., 2014*). Briefly, RAW files were converted into mzML format (ProteoWizard, v3.0.11781) (*Adusumilli and Mallick, 2017*) for input into ProTurn, along with DTAselect-filter text files for protein identification. ProTurn parameters were as follows: area-under-curve integration width: 60 p.p.m., extracted ion chromatograph smoothing: Savitzky-Golay filter (*Savitzky and Golay, 1964*) over seven data points. Non-steady state curve fitting was used to account for the initial delay in uptake of $^2$H, using a first-order kinetic curve to approximate the equilibration of $^2$H$_2$O in body water. Values of $k_p$ and $p_{ss}$ were inputted from the resultant curve fitting of $^2$H enrichment in serum samples. To control against false positive identifications, only peptides that were explicitly identified in ≥4 data points were accepted for the calculation of protein turnover. The 'Allow Peptide Modification' option was selected to include any identified post-translationally modified peptides in kinetic curve-fitting, and peptide isotopomer time series were included if $R^2$ ≥0.8 or standard error of estimate (S.E)≤0.05 (*Lam et al., 2014*). Protein-level turnover rate is reported as the median and median absolute deviation of the turnover rate constants of each accepted unique constituent peptide, from which protein half-life can be calculated assuming a first order reaction (*Lau et al., 2018*).

For some proteins of interest, particularly in the IFM, automated calculation of turnover rates was unsuccessful, due to unsuccessful automated identification of isotopomer peaks. In these cases, turnover rate constants were calculated manually, using isotopomer peak height to calculate the relative abundance of $M_0$ as a function of time (Thermo Xcalibur v2.2), and fitting first order kinetic curves to estimate $k$ (GraphPad Prism v8.0.2). As the labelling pattern of each peptide is dependent on the number of available labelling sites ($N$), unlabelled relative abundance of the 0$^{th}$ isotopomer ($a$), as well as the plateau level of enrichment of deuterium ($p_{ss}$); these values were used to calculate the plateau values of the 0$^{th}$ isotopomer ($A_0$) which occurs when the peptide is fully labelled. $N$ was calculated from the literature (*Lam et al., 2014*; *Commerford et al., 1983*). $a$ was calculated from the peptide sequence and natural abundance of heavy isotopes of carbon, nitrogen, oxygen and sulphur (*Lam et al., 2014*). $p_{ss}$ was calculated from the serum enrichment as detailed above. The following equation was then used to calculate the plateau of $A_0$:

$$A_{0,PLATEA} = a(1 - p_{ss})^N$$

Previous work demonstrates this accurately predicts the plateau measured experimentally (*Lam et al., 2014*), therefore, for curve fitting calculations the plateau was constrained to the calculated value for each peptide. To assess the degree of differences in curve fitting performed by automated and manual approaches, $k$ values were calculated using both approaches for 10 peptides. Resulting $k$ values differed by an average of 23%. This method does not allow for direct comparison between manual and automated calculations of $k$, due to differences in curve fitting method, but does allow differences in peptide $k$ values between IFM and FM to be assessed, when both are calculated manually.

## Statistical analysis

When ≥ 3 peptides were identified corresponding to a particular protein, statistical differences in turnover rate constants in tendon phases were assessed using paired t-tests, Wilcoxon matched pairs tests, or Mann-Whitney tests, with $p<0.05$ (GraphPad Prism, v8.2). The statistical test chosen was dependent on whether data were normally distributed, which was assessed with Kolmogorov-Smirnov tests, and whether the same peptides were identified in tendon phases, which allowed for paired statistical tests.

## Acknowledgements

The authors would like to thank Professor Rob Beynon for his advice regarding study design and when preparing the manuscript.

## Additional information

### Funding

| Funder | Grant reference number | Author |
|---|---|---|
| Arthritis Research UK | 21216 | Chavaunne T Thorpe |
| National Heart, Lung, and Blood Institute | R35 HL135772 | Howard Choi<br>Ding Wang<br>Peipei Ping |
| National Heart, Lung, and Blood Institute | R01 HL146739 | Howard Choi<br>Ding Wang<br>Peipei Ping |
| National Heart, Lung, and Blood Institute | T32 HL139450 | Howard Choi<br>Ding Wang<br>Peipei Ping |

The funders had no role in study design, data collection and interpretation, or the decision to submit the work for publication.

### Author contributions

Howard Choi, Software, Formal analysis, Writing - review and editing; Deborah Simpson, Investigation, Methodology, Writing - review and editing; Ding Wang, Software, Writing - review and editing; Mark Prescott, Investigation, Methodology; Andrew A Pitsillides, Jayesh Dudhia, Peter D Clegg, Conceptualization, Writing - review and editing; Peipei Ping, Resources, Software; Chavaunne T Thorpe, Conceptualization, Funding acquisition, Investigation, Methodology, Writing - original draft, Project administration, Writing - review and editing

### Author ORCIDs

Chavaunne T Thorpe ⓘ https://orcid.org/0000-0001-7051-3504

### Ethics

Animal experimentation: All procedures complied with the Animals (Scientific Procedures) Act 1986, were approved by the local ethics committee at the Royal Veterinary College, were performed under project licence PB78F43EE and are reported according to the ARRIVE guidelines. All of the animals were handled according to National Centre for the Replacement, Refinement and Reduction of Animals in Research (NC3Rs) guidelines.

### Decision letter and Author response

Decision letter https://doi.org/10.7554/eLife.55262.sa1
Author response https://doi.org/10.7554/eLife.55262.sa2

## Additional files

### Supplementary files

• Source data 1. Source data for *Figures 5* and *6*. ProTurn output for whole tendon. hl tab contains protein half-life information organized by peptide sequence and hl-data tab contains isotopomer relative abundance at each time point

• Transparent reporting form

## Data availability

The proteomic data have been deposited to the ProteomeXchange Consortium via the PRIDE partner repository with the data set identifier PXD015928 and 10.6019/PXD015928. Source files for 2H enrichment data, ProTurn output files and manually calculated turnover rates are provided with the manuscript.

The following dataset was generated:

| Author(s) | Year | Dataset title | Dataset URL | Database and Identifier |
|---|---|---|---|---|
| Choi H, Simpson D, Wang D, Prescott M, Pitsillides AA, Dudhia J, Clegg PD, Ping P, Thorpe CT | 2020 | Heterogeneity of proteome dynamics between connective tissue phases of adult tendon | http://doi.org/10.6019/PXD015928 | ProteomeXchange, 10.6019/PXD015928 |

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
