## [Decision Letter]

**Acceptance summary:**

This is an interesting study that uses mass spectrometry combined with deuterium isotope labeling (via initial injection and D2O water ad lib) to gain an unbiased look at the protein composition and turnover in the interfascicular and fascicular (IFM and FM, respectively) matrix of the rat Achilles tendon. The results support the hypothesis that the proteome of the IFM is more dynamic than the FM, and point out that these data in this model system have relevance to tendinous tissue remodeling and repair.

**Decision letter after peer review:**

Thank you for submitting your work entitled "Heterogeneity of proteome dynamics between connective tissue phases of adult tendon" for consideration by *eLife*. Your article has been evaluated by Clifford Rosen as the Senior Editor, a Reviewing Editor and one peer reviewer.

There are some issues that need to be addressed before acceptance, as outlined below:

Reviewer #1:

This is an interesting study that uses mass spectrometry combined with deuterium isotope labeling (via initial injection and D2O water ad lib) to gain an unbiased look at the protein composition and turnover in the interfascicular and fascicular (IFM and FM, respectively) matrix of the rat Achilles tendon. The results support the hypothesis that the proteome of the IFM is more dynamic than the FM, and point out that these data in this model system have relevance to tendinous tissue remodeling and repair.

Overall, the results were convincing that there is a difference in IFM and FM composition and turnover. However, some concerns were noted regarding the validation of some aspects of this model system. Addressing these concerns would strengthen the paper overall.

First, there were no corroborating data for the mass spectrometric results. For example, western blotting for IFM and IF proteins would be useful not only to corroborate the MS data but additionally, to gauge the efficiency of the separation of IFM and FM tissue sections and their relative levels of enrichment for at least 1 or a few selected proteins. An alternative approach (given the low amounts of IFM material) could include immunofluorescence or immunohistochemical staining of tendon sections for representative proteins.

Second, regarding the equal representation of sex in the experiments, justification for using only female rats should be offered.

The Discussion should address whether turnover of protein post translational modifications (e.g., glycosylation) is an important feature of tendinous remodeling. Also, the Discussion would be enhanced by a concise discussion of any relevant knockout mouse phenotypes that support the importance of IFM- or FM-expressed proteins to tendinous or other tissue function.

Finally, if this was not included, provision for submission of these MS/proteomic data should be made to an appropriate public-accessible data base (e.g., PRIDE, PRoteomics IDEntifications database), consistent with Journal policies.

Reviewer #2:

This paper investigates the proteome dynamics in tendon by comparing calculated protein turnover rates in the interfascicular matrix (IFM) vs. the fascicular matrix (FM).

This is a very interesting study illustrating the different protein turnover rates in different regions of the tendon, isolated by laser capture microdissection. The methods used are highly adequate and the data interpretation is sound. The main conclusions drawn are supported by the data presented and the study is well presented in the manuscript.

In the database search (subsection “Peptide and Protein Identification”) – it seems like you did not use oxidation of proline as a variable modification, please comment. The number of collagen peptides identified would be significantly increased by using that modification and I would highly recommend a new search using this modification to see how the collagen data is affected.

---

## [Author Response]

Reviewer #1:[…] Overall, the results were convincing that there is a difference in IFM and FM composition and turnover. However, some concerns were noted regarding the validation of some aspects of this model system. Addressing these concerns would strengthen the paper overall.First, there were no corroborating data for the mass spectrometric results. For example, western blotting for IFM and IF proteins would be useful not only to corroborate the MS data but additionally, to gauge the efficiency of the separation of IFM and FM tissue sections and their relative levels of enrichment for at least 1 or a few selected proteins. An alternative approach (given the low amounts of IFM material) could include immunofluorescence or immunohistochemical staining of tendon sections for representative proteins.

Thank you for this useful suggestion. We were planning to immunolabel some tendon sections for key ECM proteins (collagen types 1 and 3, decorin, biglycan, fibromodulin) but unfortunately have not been able to do so due to lab closures as a result of COVID-19. However, distribution of these proteins within tendon is well described in the literature, and we have discussed our findings in light of these (Discussion, third and ninth paragraphs).

Second, regarding the equal representation of sex in the experiments, justification for using only female rats should be offered.

Thank you for raising this point. We used female rats to maintain consistency with a related study, for which female rats were chosen to reduce the possibility of having to singly house animals after surgery due to animals fighting (which is common in males). We have added use of a single sex as a limitation of the study (Discussion, second paragraph).

The Discussion should address whether turnover of protein post translational modifications (e.g., glycosylation) is an important feature of tendinous remodeling. Also, the Discussion would be enhanced by a concise discussion of any relevant knockout mouse phenotypes that support the importance of IFM- or FM-expressed proteins to tendinous or other tissue function.

Thank you for raising this point, which we had not considered. We have added a section to the Discussion considering the effect of PTMs on tendon remodelling (Discussion, fourth paragraph). We have also expanded on the discussion of relevant KO mice phenotypes (Discussion, ninth paragraph).

Finally, if this was not included, provision for submission of these MS/proteomic data should be made to an appropriate public-accessible data base (e.g., PRIDE, PRoteomics IDEntifications database), consistent with Journal policies.

The data have been uploaded to the PRIDE repository, as described in the subsection “Peptide and Protein Identification”.

Reviewer #2:[…] In the database search (subsection “Peptide and Protein Identification”) – it seems like you did not use oxidation of proline as a variable modification, please comment. The number of collagen peptides identified would be significantly increased by using that modification and I would highly recommend a new search using this modification to see how the collagen data is affected.

Thank you for this suggestion, we have now included oxidation of proline as a variable modification in the searches, which has resulted in a large increase number of collagen peptides identified in whole tendon, FM and IFM (Results). We have re-calculated protein half-life using this data, which has allowed us to calculate the turnover rates of a number of additional minor collagens (see Figure 6). It has also increased the number of collagen type I peptides identified, increasing the significance of our results (see Figure 7).